# Load-to-Failure Resistance and Optical Characteristics of Nano-Lithium Disilicate Ceramic after Different Aging Processes

**DOI:** 10.3390/ma15114011

**Published:** 2022-06-05

**Authors:** Mustafa Borga Donmez, Emin Orkun Olcay, Münir Demirel

**Affiliations:** 1Department of Reconstructive Dentistry and Gerodontology, School of Dental Medicine, University of Bern, 3010 Bern, Switzerland; 2Department of Prosthodontics, Faculty of Dentistry, Istinye University, 34010 İstanbul, Turkey; 3Department of Prosthodontics, Faculty of Dentistry, Biruni University, 34010 İstanbul, Turkey; eolcay@biruni.edu.tr; 4Oral and Dental Health Department, Vocational School, Biruni University, 34010 İstanbul, Turkey; munirdemirel@biruni.edu.tr

**Keywords:** color stability, load-to-failure resistance, nano-lithium disilicate, relative translucency parameter

## Abstract

The aim of this study was to compare the load-to-failure resistance and optical properties of nano-lithium disilicate (NLD) with lithium disilicate (LDS) and zirconia-reinforced lithium silicate (ZLS) in different aging processes. Thirty crowns were milled from NLD, LDS, and ZLS (n = 10). All crowns were subjected to thermomechanical aging and loaded until catastrophic failure. Ten specimens from each material were prepared in two different thicknesses (0.7 mm and 1.5 mm, n = 5), and color coordinates were measured before and after coffee thermocycling. Color differences (ΔE00) and relative translucency parameter (RTP) were calculated. Data were analyzed by using ANOVA and Bonferroni-corrected *t*-tests (α = 0.05). ZLS had the highest load-to-failure resistance (*p* ≤ 0.002), while the difference between LDS and NLD was nonsignificant (*p* = 0.776). The interaction between material type and thickness affected ΔE00 (*p* < 0.001). Among the 0.7 mm thick specimens, ZLS had the lowest ΔE00 (*p* < 0.001). Furthermore, 1.5 mm thick ZLS had lower ΔE00 than that of 1.5 mm thick LDS (*p* = 0.036). Other than ZLS (*p* = 0.078), 0.7 mm thick specimens had higher ΔE00 (*p* < 0.001). The interaction between material type, thickness, and thermocycling affected RTP (*p* < 0.001). Thinner specimens presented higher RTP (*p* < 0.001). NLD and LDS had higher RTP than ZLS (*p* ≤ 0.036). However, 0.7 mm thick specimens had similar RTP after coffee thermocycling (*p* ≥ 0.265). Coffee thermocycling reduced the RTP values of 0.7 mm thick NLD (*p* = 0.032) and LDS (*p* = 0.008). NLD may endure the occlusal forces present in the posterior region. However, long-term coffee consumption may impair the esthetics of restorations particularly when thin NLD is used.

## 1. Introduction

With the recent advancements in computer-aided design–computer-aided manufacturing (CAD-CAM) technologies [1,2], ceramics can now be used in monolithic forms [3]. In addition, a wide range of CAD-CAM ceramics with different crystalline structures [4,5,6] are now available for various restorative options [1]. Among these materials, lithium silicate ceramics are widely preferred for prosthetic treatments [7]. 

A well-known example of lithium silicate ceramics is lithium disilicate glass-ceramic (LDS; IPS e.max CAD, Ivoclar Vivadent, Schaan, Liechtenstein), which has been in the dental market since 2006 [8]. LDS gained attraction due to its exceptional esthetics [9] and superior physical properties, compared with conventional silicate ceramics [10]. However, new materials that have similarities and differences with LDS are frequently introduced [11,12]. One of those materials was zirconia-reinforced lithium silicate glass-ceramic (ZLS; Vita Suprinity, Vita Zahnfabrick, Bad Säckingen, Germany), which was launched as a unique material that combines the advantages of lithium disilicate and zirconia [13,14] due to 10 wt% of zirconium dioxide dissolved in its glassy matrix [15]. Recently, a new lithium disilicate called nano-lithium disilicate glass-ceramic (NLD; Amber Mill, Hass, Gangneung, Korea) has been marketed [7,8,12]. NLD differs from currently available lithium silicate ceramics, as its translucency is adjustable depending on the crystallization process [16]. Even though NLD has started to gain attention, the number of studies on this material is scarce [7,12,16,17,18,19,20].

The success of restoration depends on the mechanical and optical properties of the material [21]. An ideal restorative material must withstand the occlusal forces [22] while also matching the optical properties of the natural teeth [23]. However, there is an inverse relationship between the mechanical and the esthetic properties as increasing the crystalline content to achieve greater strength comprises optical properties [5]. As for the esthetic outcomes, translucency is a key component [24], which is related to several factors [25] including material thickness [24]. Color stability of restoration is also critical for its longevity [26,27] and aging may affect the optical characteristics [28]. There are studies focusing on the optical properties of lithium silicate ceramics after thermal aging [11,26,27,28,29]. However, considering the chemical differences among different lithium silicate glass-ceramics [7], clinicians would benefit from studies that report performance differences among these materials. In addition, the existing studies on NLD [7,12,16,17,18,19,20] did not investigate its fracture resistance, and only three studies have focused on its optical properties, which did not involve aging [12,16,19]. Therefore, this study aimed to compare NLD with LDS and ZLS in terms of load-to-failure resistance, color stability, and translucency after aging. The null hypotheses were that (1) material type would not affect the load-to-failure resistance, (2) material type and thickness would not affect the color stability, and (3) material type, material thickness, and coffee thermocycling would not affect the translucency.

## 2. Materials and Methods

### 2.1. Crown Fabrication and Load-to-Failure Resistance Test

Table 1 lists the CAD-CAM materials used in the present study. The number of specimens in each group was set as 10 for the load-to-failure resistance test [30,31,32,33,34] and as 5 for color measurements [23,27,28,35,36], based on previous studies that reported significant differences.

A mandibular cast with a prepared left first molar for an all-ceramic crown (Figure 1) was digitized by using a laboratory scanner (inEos X5, Dentsply Sirona, Bensheim, Germany), and a complete-coverage crown with an occlusal thickness of 1.5 mm was designed in standard tessellation language (STL) format. Thirty reinforced composite resin dies (BRILLIANT Crios, Coltène AG, Altstätten, Switzerland) and their corresponding crowns (LDS, NLD, and ZLS) were milled with a milling unit (CEREC MC XL, Dentsply Sirona, Bensheim, Germany). Thereafter, all crowns were crystallized according to their respective manufacturers’ recommendations (Table 2) by using a porcelain furnace (Programat P310, Ivoclar Vivadent, Schaan, Liechtenstein). 

Prior to cementation, the dies were sandblasted using 50 μm aluminum oxide and treated with a universal adhesive (Single Bond Universal Adhesive, 3M ESPE, St. Paul, MN, USA). Intaglio surfaces of the crowns were etched with 5% hydrofluoric acid (IPS Ceramic Etching Gel, Ivoclar Vivadent, Schaan, Liechtenstein) (20 s for LDS and ZLS, 30 s for NLD), water-rinsed, and air-dried. A thin coat of universal primer (Monobond Plus, Ivoclar Vivadent, Schaan, Liechtenstein) was applied to the intaglio surfaces with a microbrush for 60 s, and any remaining excess was dispersed with a stream of air. A dual-cure resin cement (Variolink DC, Ivoclar Vivadent, Schaan, Liechtenstein) was used to cement all crowns with finger pressure. An LED-curing unit (Bluephase, Ivoclar Vivadent, Schaan, Liechtenstein) with a light intensity of 1200 mW/cm^2^ was applied for 30 s on each surface, and specimens were then stored in distilled water (37 °C) for 24 h [22].

All specimens were embedded in prefabricated plastic molds 3 mm below the cervical line by using an auto-polymerizing acrylic resin (Meliodent, Bayer Dental Ltd., Newbury, UK). After fixing the molds to the lower part of the mastication simulator, the specimens were subjected to a 49 N load for 1.2 × 10^5^ cycles, with a frequency of 1.2 Hz, by using a 6 mm stainless steel ball (Mastication Simulator, Esetron Smart Robotechnologies, Ankara, Turkey). Simultaneous thermocycling was performed for 5000 cycles in 5 °C and 55 °C water, with a dwell time of 60 s and transfer time of 10 s. These parameters were reported to simulate approximately 5 years of clinical service [10]. All crowns were then analyzed by using a stereomicroscope (Olympus SZ61, Olympus Corporation, Tokyo, Japan) at 40× magnification for the presence of any cracks. After that, a load-to-failure resistance test was performed with a universal testing machine (Lloyd LRX, Lloyd Instruments, West Sussex, UK). The load was vertically applied to the central occlusal fossa of the crowns until catastrophic fracture by using a 6 mm diameter stainless steel ball with a crosshead speed of 0.5 mm/min. An even distribution of forces was accomplished by placing a thin thermoplastic film between the loading tip and the crown [33,37]. The maximum load at failure was automatically recorded in Newton (N).

### 2.2. Color Coordinate Measurements

Thirty rectangle-shaped specimens with two different thicknesses (0.7 mm and 1.5 mm, A2 shade) were prepared by using a low-speed precision cutter (Micracut 151, Metkon, Bursa, Turkey) under water cooling and crystallized according to their respective manufacturers’ recommendations (n = 5 for each subgroup). NLD specimens were crystallized to achieve high translucency (Closing temperature: 400 °C, heat rate: 60 °C/min, final temperature: 815 °C/min, fusion time: 15 min) [17], and other materials were prepared from high translucent blocks. All specimens were then polished by using #600-1200 silicon carbide abrasive papers. Final thicknesses were controlled by using a digital caliper (Absolute Digimatic, Mitutoyo, Tokyo, Japan), and all specimens were cleaned in an ultrasonic bath with distilled water (Biosonic UC 50DB, Coltene Whaledent, Cuyahoga Falls, OH, USA) for 10 min.

The baseline color coordinates were measured on black, gray, and white backgrounds by using a spectrophotometer (CM-26d, Konica Minolta, Tokyo, Japan), which uses the CIE Standard (2°) human observer characteristics. The spectrophotometer was calibrated per the manufacturer’s guidelines before each measurement. The same practitioner (M.B.D.) repeated the measurements for each specimen 3 times before calculating the mean values of each specimen. Following the baseline measurements, specimens were subjected to 5000 cycles of coffee thermocycling between 5 °C and 55 °C, with a dwell time of 30 s (Thermocycler THE 1100, SD Mechatronik, Feldkirchen-Westerham, Germany), as previously described [24,26,27,29]. After coffee thermocycling, coffee stains were removed by gently brushing the specimens 10 times with toothpaste (Colgate Total Pro Breath Health, Colgate-Palmolive, New York, NY, USA) under running water. Specimens were then ultrasonically cleaned for 15 min and dried with tissue paper; color measurements were repeated similar to the baseline measurements. 

The color differences in the specimens between baseline and after coffee thermocycling were calculated by using the coordinates measured on a gray background. CIEDE2000 (ΔE_00_) color difference formula was used for the calculations, in which parametric factors kL, kC, and kH were set as 1 [26,29].
ΔE_00_ = [(ΔL′/k_L_S_L_)^2^ + (ΔC′/k_C_S_C_)^2^ + (ΔH′/k_H_S_H_)^2^ +R_T_(ΔC′/k_C_S_C_)(ΔH′/k_H_S_H_)]^1/2^

The color coordinates measured on black (L* = 8.2, a* = −0.3, b* = 1.44) and white (L* = 90.4, a* = −2.2, b* = 1.3) backgrounds were used for the relative translucency parameter (RTP) calculations of the specimens both for baseline and after coffee thermocycling [26,29].

One additional specimen from each material was etched using 5% hydrofluoric acid, as mentioned before, and examined by using a scanning electron microscope (EVO LS-10, Zeiss, Cambridge, UK) at 5000× magnification, to observe the microtopography and the crystalline structure.

### 2.3. Statistical Analysis

A statistical analysis software (SPSS/PC Version 23.0, 2021, SPSS Inc., Chicago, IL, USA) was used to analyze the data. Statistical differences in load-to-failure resistance values of the materials were determined by using 1-way analysis of variance (ANOVA) and Tukey HSD tests. Color differences caused by coffee thermocycling were analyzed by using the color coordinates measured over the gray background before and after aging. Mean ΔE_00_ values and 95% confidence limits were calculated for each type of material and thickness. Data were analyzed with a 2-way ANOVA with material type and thickness being the main effects, and the interaction was included. Mean RTP values and 95% confidence limits were calculated for material type, thickness, and aging condition. Repeated measures 3-way ANOVA was used to evaluate RTP values with material type and thickness as between-subject factors, and aging condition being the within-subject factor, including the interactions. Any significant interaction was further resolved by using Student’s *t*-test with Bonferroni correction (α = 0.05) [29]. 

## 3. Results

All crowns survived thermomechanical aging and were further subjected to a load-to-failure resistance test. One-way ANOVA revealed that tested materials had significantly different load-to-failure resistance (F = 11.59, df = 2, and *p* < 0.001). As Figure 2 shows, ZLS crowns had higher Load-to-failure resistance values than LDS (*p* < 0.001; the estimated difference in means: 141 N) and NLD (*p =* 0.002, the estimated difference in means: 119.5 N), whereas the difference between LDS and NLD was nonsignificant (*p* = 0.776).

Table 3 presents the two- and three-way ANOVA results of the optical parameters. Material type, thickness, and their interactions had a significant effect on ΔE_00_ (*p* < 0.001). The difference between 0.7 mm thick NLD and LDS was nonsignificant (*p* > 0.05), whereas 0.7 mm thick ZLS had the lowest ΔE_00_ (*p* < 0.001). Among the 1.5 mm thick specimens, LDS had higher ΔE_00_ than ZLS (*p* = 0.036), while NLD had similar values to those of LDS (*p* < 0.001) and ZLS (*p* < 0.001). Furthermore, 0.7 mm thick specimens showed a greater color change than 1.5 mm thick specimens (*p* < 0.001), except for ZLS (*p* = 0.078) (Figure 3). 

Material type, thickness, and coffee thermocycling (*p* < 0.001), as well as every possible interaction among the main effects, affected RTP values (*p* ≤ 0.035). Significant interactions of the clinically relevant comparisons were resolved by using the same material of different thicknesses and different materials of the same thickness (Table 4). Within each material, 0.7 mm thick specimens had higher RTP regardless of the coffee thermocycling (*p* < 0.001). In addition, LDS and NLD had similar RTP values regardless of the thickness and coffee thermocycling (*p ≥* 0.265). Other than 0.7 mm thick specimens after coffee thermocycling (*p ≥* 0.265), RTP values of ZLS were significantly lower than those of NLD and LDS (*p* ≤ 0.036). Coffee thermocycling significantly reduced the RTP of 0.7 mm thick NLD (*p* = 0.032) and 0.7 mm thick LDS (*p* = 0.008) (Figure 4).

## 4. Discussion

The first null hypothesis was rejected, as the differences among the load-to-failure resistance values of tested materials were significant. Lubauer et al. [7] showed that ZLS and LDS consisted of lithium metasilicate prior to crystallization, and the crystallization process reduced the glassy phase of all materials tested in the present study. After crystallization, ZLS, which had the highest load-to-failure resistance values in the present study, was reported to comprise 29 vol% lithium metasilicate, 13.7 vol% lithium disilicate, 13.3 vol% lithium orthophosphate, and 42.9 vol% glass. However, NLD (62.6 vol% lithium disilicate, 7.2 vol% lithium orthophosphate, and 29.7 vol% glass) and LDS (46.1 vol% lithium disilicate, 6.2 vol% lithium orthophosphate, and 33.7 vol% glass) were reported not to comprise any lithium metasilicate after crystallization, while quartz (13.2 vol%) was present in NLD [7]. The authors of [7] have also concluded that, after crystallization, both base (4.52 mol%) and residual (14.1 mol%) glass of ZLS contained higher amounts of zirconium dioxide than those of NLD (base: 0.009 mol% and residual: 0.034 mol%) and LDS (base: 0.375 mol% and residual: 1.29 mol%), which led to an increased network polymerization during crystallization. This increased polymerization may be attributed to the higher load-to-failure resistance values of ZLS. The authors believe that the present study is the first on the fracture resistance of NLD; thus, a comparison with previous studies was not possible. Nevertheless, ZLS had higher load-to-failure resistance values than LDS, which is in line with previous studies [6,38,39]. Nevertheless, there are also studies showing that LDS had load-to-failure resistance that was either higher than [1,10] or similar to [30,31,32,40] ZLS. Considering these varying results, future studies are needed to corroborate the findings of the present study. Still, given that occlusal forces alternate from 600 N to 800 N in the posterior region [40,41], or even exceed these values for bruxer patients [42], the load-to-failure resistance values reached in this study (1135 N to 1341 N for LDS, 1173 N to 1395 N for NLD, and 1244 N to 1484 N for ZLS) were satisfactory for all materials tested. 

Previous studies have investigated NLD’s different mechanical properties [7,17,18,20]. Yin et al. [20] have reported that NLD had higher Vickers hardness and biaxial flexural strength values, along with a more stable surface topography than CAD-CAM composite blocks. In another study, NLD was shown to have similar biaxial flexural strength with LDS and a different brand of ZLS after hydrothermal aging [17]. A recent study has investigated the fracture toughness of lithium silicate ceramics and concluded that NLD had higher values than ZLS [7]. Kang et al. [18] showed that NLD and LDS have similar milling accuracy that was higher than that of ZLS. Considering the results of the present and those previous studies on NLD, it can be speculated that NLD might be a favorable material mechanically, but this hypothesis should be substantiated with future in vivo studies.

Load-to-failure resistance of all-ceramic crowns primarily depends on restoration design, tooth preparation, cementation, and material thickness. While combining these four factors enhances the load-to-failure resistance, material type and thickness emerge as key components [40]. The present study solely focused on the comparison of the load-to-failure resistance of NLD with LDS and ZLS after thermomechanical loading; thus, identical crown and die designs were used. In addition, finger pressure was used during cementation due to the fact that this method is commonly used by clinicians. 

The second null hypothesis was rejected, as material type and thickness affected color stability. In the present study, perceptibility and acceptability thresholds were set as 0.8 units and 1.8 units [43]. ΔE_00_ values of 0.7 mm thick NLD and LDS were similar to each other and were higher than the perceptibility threshold. However, every other material–thickness pair had imperceptible color change (Figure 3). A previous study investigated the effect of resin cement shade on the color stability of CAD-CAM ceramics including NLD and LDS [19]. The authors of [19] have concluded that NLD had color stability either similar to or higher than LDS, while the color change in both materials was unacceptable regardless of the resin cement shade. In another study, LDS was found to maintain its shade after required and additional crystallization, whereas NLD differed from its original shade even after the first crystallization firing [16]. Considering the novelty of NLD and the fact that the present study is the first to address the comparison of NLD and ZLS in terms of color stability, these findings need to be substantiated. 

The translucency of the specimens was significantly affected by material type, thickness, and coffee thermocycling. Therefore, the third null hypothesis was rejected. RTP values of 0.7 mm thick specimens were significantly higher regardless of the material type and coffee thermocycling. LDS and NLD presented higher RTP values than ZLS except for the 0.7 mm thick specimens after coffee thermocycling, as coffee thermocycling reduced the translucency of 0.7 mm thick LDS and NLD specimens (Figure 4). These results comply with previous studies [28,29]. In a recent study by Salas et al. [44], translucency perceptibility and acceptability thresholds were defined as 0.62 and 2.62 units. Among the material-thickness pairs tested, 0.7 mm thick NLD (ΔRTP = 5.27), and 0.7 mm thick LDS (ΔRTP = 5.52) had unacceptable RTP changes, while 1.5 mm thick LDS (ΔRTP = 1.51) and 1.5 mm thick ZLS (ΔRTP = 1.09) had perceptible RTP changes.

The type, form, and volume of the crystalline structure of a ceramic affect its translucency [25]. Even though SEM images (Figure 5) showed that NLD and ZLS had similar crystalline sizes that were smaller than that of LDS [7], the denser microcrystalline structure of ZLS may be associated with its significantly lower baseline RTP values. However, these results may change according to the different translucency levels, considering that the present study only evaluated high translucent specimens. Moreover, considering NLD’s unique adjustable translucency feature, future studies investigating this aspect, as well as the possible effects of re-firing on the translucency of NLD, might elaborate the understanding of this concept. 

A limitation of this in vitro study was the constant thickness of the specimens used for the load-to-failure resistance test. Previous studies have shown statistically lower load-to-failure resistance values for restorations with reduced occlusal thickness while testing LDS and ZLS [10,32,40]. Aging methods might also be considered as limitations. Thermomechanical aging was performed by using distilled water instead of artificial saliva, while both sides of the specimens were subjected to coffee thermocycling. Intaglio surfaces of restorations are not exposed to staining solutions clinically; thus, a more evident color change might have been observed in the present study [29]. In addition, the effects of surface treatments (glazing or polishing) [26] and resin cements [11,19] on the color change in CAD-CAM materials have been reported. Future studies involving these components with diversified test parameters to other features affecting the success of restoration can enlighten the clinical outcomes of NLD.

## 5. Conclusions

Considering the limitations of the present study, it can be concluded that NLD may have pleasing clinical performance for complete-coverage restorations at the molar region. In addition, NLD may have esthetic outcomes similar to LDS and better than ZLS. However, clinicians should consider the effect of long-term coffee consumption on the color stability of NLD, particularly for those restorations with reduced thicknesses. 

## Figures and Tables

**Figure 1 materials-15-04011-f001:**
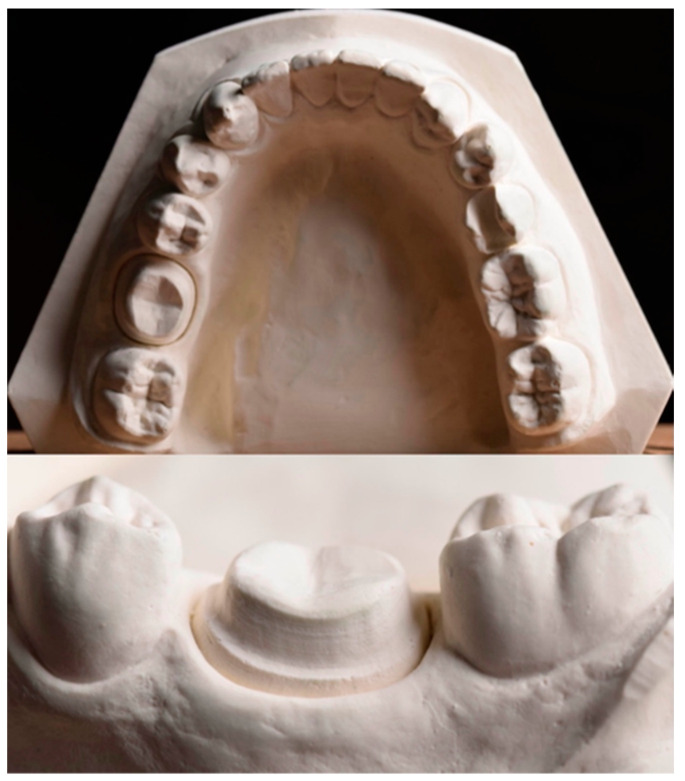
Reference cast.

**Figure 2 materials-15-04011-f002:**
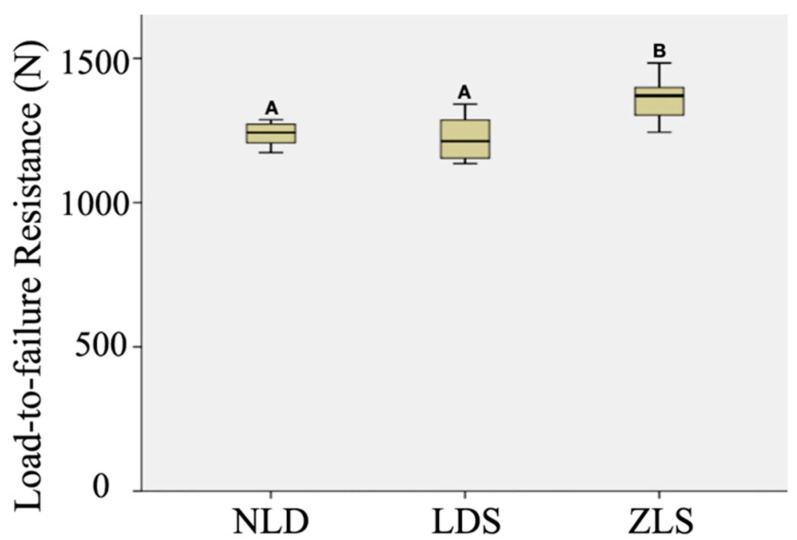
Box plot graph of load-to-failure resistance of all materials; different uppercase letters present significant differences among groups, *p* < 0.05.

**Figure 3 materials-15-04011-f003:**
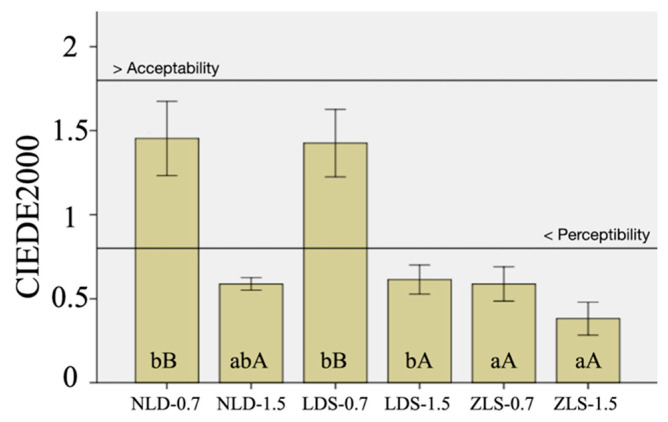
Means and 95% confidence limits of ΔE00 values for each material–thickness pair. Different lowercase letters indicate significant differences among materials with the same thickness, while uppercase letters indicate significant differences between different thicknesses of the same material (*p* < 0.05). Horizontal lines represent the perceptibility threshold (0.8 units) and the acceptability threshold (1.8 units).

**Figure 4 materials-15-04011-f004:**
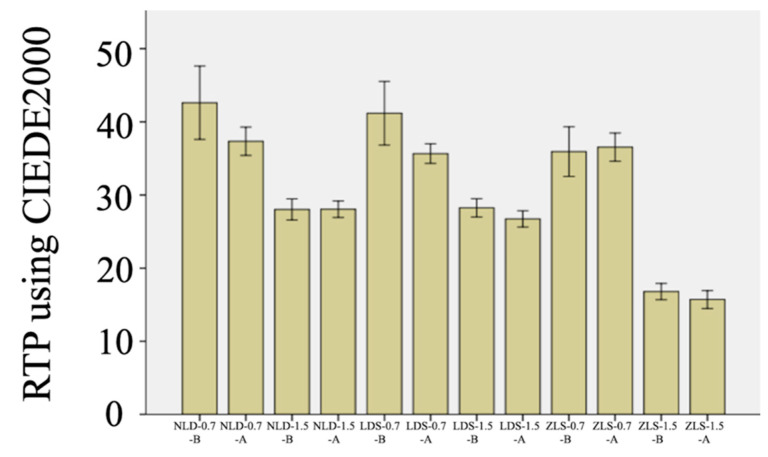
Mean RTP values and 95% confidence limits for each material–thickness pair: B, baseline; A, after coffee thermocycling.

**Figure 5 materials-15-04011-f005:**

SEM images of tested materials after 5% hydrofluoric acid etching: (**A**) NLD; (**B**) LDS; (**C**) ZLS.

**Table 1 materials-15-04011-t001:** List of CAD-CAM materials used in this study.

Material	Classification	Manufacturer
Amber Mill	Nano-lithium disilicate glass-ceramic, NLD	Hass, Gangneung, Korea
IPS e.max CAD	Lithium disilicate glass-ceramic, LDS	Ivoclar Vivadent, Schaan, Liechtenstein
Vita Suprinity	Zirconia-reinforced lithium silicate glass-ceramic, ZLS	Vita Zahnfabrick, Bad Säckingen, Germany
BRILLIANT Crios	Reinforced composite resin	Coltène AG, Altstätten, Switzerland

**Table 2 materials-15-04011-t002:** Crystallization parameters of the materials used in the present study.

	B (°C)	S (min)	t1/t2 (°C/min)	T1/T2 (°C)	H1/H2 (min)	Vac. 1 (°C)/Vac. 2 (°C)	L (°C)	tL
Amber Mill	400 °C	3 min	60 °C/min	815 °C	15 min	550/815 °C	690 °C	0
IPS e.max CAD	403 °C	6 min	90/34 °C/min	830/850 °C	10 s–7 min	550–830/830–850 °C	710 °C	0
Vita Suprinity	400 °C	4 min	55 °C/min	840 °C	8 min	410/839 °C	680 °C	0

**Table 3 materials-15-04011-t003:** ANOVA results of the color parameters.

Color Change	Sum of Squares	df	Mean Square	F	*p*
Material	1.912	2	0.956	74.726	<0.001
Thickness	2.958	1	2.958	231.235	<0.001
Material × Thickness	0.671	2	0.336	26.246	<0.001
**RTP**	**Sum of Squares**	**df**	**Mean Square**	**F**	** *p* **
Material	710.535	2	355.268	89.110	<0.001
Thickness	3061.061	1	3061.061	767.793	<0.001
Condition	67.628	1	67.628	16.963	<0.001
Material × Thickness	246.614	2	123.307	30.929	<0.001
Material × Condition	28.675	2	14.337	3.596	0.035
Thickness × Condition	24.219	1	24.219	6.075	0.017
Material × Thickness × Condition	34.709	2	17.354	4.353	0.018

R-squared = 0.948 (adjusted R-squared = 0.937); R-squared = 0.956 (adjusted R-squared = 0.946).

**Table 4 materials-15-04011-t004:** Comparison of *p* values for RTP values of clinically relevant pairs at the baseline and after coffee.

Pairs	Baseline	After Coffee Thermocycling
0.7 NLD-0.7 LDS	0.946	0.265
0.7 NLD-0.7 ZLS	0.004	0.892
0.7 NLD-1.5 NLD	<0.001	<0.001
0.7 LDS-0.7 ZLS	0.036	0.85
0.7 LDS-1.5 LDS	<0.001	<0.001
0.7 ZLS-1.5 ZLS	<0.001	<0.001
1.5 NLD-1.5 LDS	>0.05	0.517
1.5 NLD-1.5 ZLS	<0.001	<0.001
1.5 LDS-1.5 ZLS	<0.001	<0.001

Coffee thermocycling. *p* < 0.05 indicates significant differences between groups: NLD, nano-lithium disilicate; LDS, lithium disilicate; ZLS, zirconia reinforced lithium silicate.

## Data Availability

Data sharing is not applicable for this paper.

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
