# Peer review of "Load-to-Failure Resistance and Optical Characteristics of Nano-Lithium Disilicate Ceramic after Different Aging Processes"

_materials, 2022, doi:10.3390/ma15114011_

Round 1
Reviewer 1 Report
Dear Authors
Abstract: is very extensive and too descriptive.
Keywords: Not mentioned.
Introdcution:
1) In introduction add information from this paper (https://www.sciencedirect.com/science/article/pii/B9780081024768000013)
2) add some papers related to core of the topic and increase the recent information.
3) Result looks not well explained. Authors need to use proper statements and good flow for readers. If possible just use help from expert of the subject for improving the whole heading.
5) Conclusion: authors have to show future direction in clinical dentistry from the outcomes they observed in their work.
Author Response
Reviewer #1
Response: The authors would like to thank Reviewer #1 for their comments and contributions to the scientific quality of our paper. A point-by-point response to the Reviewer’s comments is below. We believe that the revisions prompted by these comments have strengthened our manuscript.
Abstract: is very extensive and too descriptive.
Response: Abstract is revised for conciseness and improved readability.
Keywords: Not mentioned.
Response: Keywords are now given at the end of the Abstract.
Introduction:
1) In introduction add information from this paper (https://www.sciencedirect.com/science/article/pii/B9780081024768000013)
Response: Mentioned paper is now referred in Introduction.
2) Add some papers related to core of the topic and increase the recent information.
Response: New references on that are related to nano-lithium disilicate are now added. In addition, Introduction is now elaborated with recent information.
3) Result looks not well explained. Authors need to use proper statements and good flow for readers. If possible just use help from expert of the subject for improving the whole heading.
Response: Results section is revised to improve flow and clarity.
5) Conclusion: authors have to show future direction in clinical dentistry from the outcomes they observed in their work.
Response: Conclusion section is revised and now reads “Considering the limitation of the present study, it can be concluded that NLD may have pleasing clinical performance for complete-coverage restorations at the molar region. In addition, NLD may have esthetic outcomes similar to LDS and better than ZLS. However, clinicians should consider the effect of long-term coffee consumption on the color stability of NLD, particularly for those restorations with reduced thicknesses.”.

Reviewer 2 Report
In the article “Fracture resistance and optical characteristics of nano-lithium disilicate ceramic after aging: an in vitro study” the compare studies of the fracture resistance and optical properties of various types of glass ceramics containing lithium silicate and zirconium oxide are shown. These results are very interesting, however some changes are needed to improve the manuscript.
1. What did the authors mean by the process of the ceramic “aging”?
2. “In vitro study” this expression is usually applied to describe biomedical research in which culture media, cell cultures, bacteria are used. Such studies were not carried out in this work.
3. The authors argue that the type of material has a significant impact on fracture resistance and optical parameters. All materials described in the article are glass-ceramic, i.e. they contain a glass phase and crystalline phase of lithium silicate. What is the ratio of these phases in each specific material? It is necessary to present the results of XRD analysis.
4. ZLS ceramic contains ZrO2. What is its modification and quantity?
5. Why did the authors use the crystallization process and how did the phase composition of glass-ceramic materials changed?
Author Response
Reviewer 2
In the article “Fracture resistance and optical characteristics of nano-lithium disilicate ceramic after aging: an in vitro study” the compare studies of the fracture resistance and optical properties of various types of glass ceramics containing lithium silicate and zirconium oxide are shown. These results are very interesting; however, some changes are needed to improve the manuscript.
Response: The authors would like to thank Reviewer #2 for their comments and contributions to the scientific quality of our paper. A point-by-point response to the Reviewer’s comments is below. We believe that the revisions prompted by these comments have strengthened our manuscript.
- What did the authors mean by the process of the ceramic “aging”?
Response: Present study investigated mechanical and optical properties of nano-lithium disilicate after either thermomechanical aging or coffee thermocycling. The authors believe that stating both methods in the title would not be appropriate. However, title of the manuscript is revised for clarity and now reads “Load-to-failure resistance and optical characteristics of nano-lithium disilicate ceramic after different aging processes”.
- “In vitro study” this expression is usually applied to describe biomedical research in which culture media, cell cultures, bacteria are used. Such studies were not carried out in this work.
Response: Title of the manuscript is revised for clarity and now reads “Load-to-failure resistance and optical characteristics of nano-lithium disilicate ceramic after different aging processes”.
- The authors argue that the type of material has a significant impact on fracture resistance and optical parameters. All materials described in the article are glass-ceramic, i.e. they contain a glass phase and crystalline phase of lithium silicate. What is the ratio of these phases in each specific material? It is necessary to present the results of XRD analysis.
Response: Even though the present study did not perform a phase quantification analysis by using XRD, a previous study on modern lithium silicate glass-ceramics (Reference #7) has focused on this aspect. Results of this study are now highlighted in Introduction to emphasize the need of comparative studies on different lithium silicate glass-ceramics as well as in Discussion to elaborate the findings of the present study.
- ZLS ceramic contains ZrO2. What is its modification and quantity?
Response: The present study did not investigate the chemical composition of tested materials. However, results of a previous study (Reference #7) are now given in the first paragraph of Discussion to elaborate the quantity of zirconium dioxide not only in ZLS but also in LDS and NLD after crystallization as well as its role on crystallization.
- Why did the authors use the crystallization process and how did the phase composition of glass-ceramic materials changed?
Response: Each material was crystallized according to its respective manufacturer’s recommendations as this is the workflow that is used in clinical situations. In addition, a previous study (Reference #7) has shown that crystallization of tested materials reduced the glassy phase and increased the fractions of different crystals, which is now mentioned in the first paragraph of Discussion.
